# Cannabinoids—Perspectives for Individual Treatment in Selected Patients: Analysis of the Case Series

**DOI:** 10.3390/biomedicines10081862

**Published:** 2022-08-02

**Authors:** Michał Graczyk, Agata Anna Lewandowska, Piotr Melnyczok, Adam Zgliński, Małgorzata Łukowicz

**Affiliations:** 1Department of Palliative Care, Collegium Medicum in Bydgoszcz, Nicolaus Copernicus University in Toruń, 87-100 Bydgoszcz, Poland; 210th Military Research Hospital and Polyclinic in Bydgoszcz, 85-681 Bydgoszcz, Poland; melnyczokpiotr@gmail.com; 3Department of Internal Medicine, District Hospital in Golub-Dobrzyń, 87-400 Golub-Dobrzyn, Poland; adam.zglinski@gmail.com; 4Department of Rehabilitation, Center of Postgraduate Medical Education, Gruca Orthopedic and Trauma Teaching Hospital in Otwock, 05-400 Otwock, Poland; gosialukowicz@wp.pl

**Keywords:** cannabis, cannabinoids, perspectives, CBD, THC, opioids

## Abstract

Cannabinoids can be successfully used in the treatment of many symptoms and diseases; however, most often they are not the drugs of first choice. They can be added to the primary therapy, which can improve its effectiveness, or be introduced as the basic treatment when the conventional methods have failed. Small clinical trials and case reports prove the benefits of applying medicinal cannabis in various indications; however, clinical trials in larger groups of patients are scarce and often controversial. Due to limited scientific evidence, it is essential to conduct further experimental trials. Understanding the role of endocannabinoids, as well as the composition of cannabis containing both phytocannabinoids and terpenes plays an important role in their clinical use. The clinical effects of cannabinoids depend, among other things, on the activity of the endocannabinoid system, the proportion of phytocannabinoids, such as Δ9-tetrahydrocannabinol (THC) and cannabidiol (CBD), and the dosage used. The article discusses the role of phytocannabinoids and the potential of using them in different clinical cases in patients suffering from chronic pain, opioid dependence, depression and migraine, who did not respond to the conventional therapeutic methods. In each of the presented cases, the implementation of cannabinoids altered the course of the disease and resulted in symptom relief. Every decision to introduce cannabinoids to the treatment should be made individually with careful attention paid to details. Additionally, it is worth taking care of good clinical communication and education so that the implemented therapy is safe, effective and properly perceived by the patient.

## 1. Introduction

Recently there has been a growing interest in the usage of medicinal cannabinoids, especially for difficult-to-treat symptom control in people who suffer from advanced diseases. The biological effects of cannabinoids are mediated by G protein-coupled cannabinoid 1 and 2 receptors (CB1, CB2), which are widely located not only in the nervous system, but also in different visceral organs, skeletal muscles, skin and adipose tissue [1]. CB1 receptor is predominantly expressed in the central nervous system (CNS) [2], such as the amygdala, hippocampus, cerebral cortex, basal ganglia and cerebellum [3,4,5]. CB2 receptors, responsible for the anti-inflammatory effect of cannabinoids, are located majorly in immune cells [1,6], but also within astrocytes and microglia, where they are involved in the modulation of the immune response, cell migration and cytokine release [5]. What is more, cannabinoids interact with some non-cannabinoid receptors, including vanilloid receptor 1 (TRPV1) [7], transient receptor potential ankyrin 1 (TRPA1) [8], G55 protein-coupled receptor (GPR55) [9] and peroxisome proliferator-activated receptors (PPARα, PPARγ) [4,10]. The diversity of cannabinoids’ mechanisms of action might partially explain their pharmacological effects in very different clinical situations. However, in spite of interesting therapeutic potential, there are still many controversies and questions to be answered related to the usage of cannabinoids in medical practice.

Δ9-tetrahydrocannabinol (THC) and cannabidiol (CBD) are definitely the most recognized phytocannabinoids, derived most often from *Cannabis sativa* L. and *Cannabis indica* L., which are used in the majority of clinical trials conducted to date [11]. While THC is mainly responsible for side effects, such as psychosis, intoxication, anxiety and sedation, its therapeutic potential involves analgesia, muscle relaxation and decrease in nausea. Despite their reputation in the society, apart from THC, phytocannabinoids present minimal psychoactive activity. Effects of CBD mostly depend on the stimulation of neurotransmission mediated by the serotonin 1A receptor (5-HT_1A_) [12,13] and on decrease in metabolism and reuptake of anandamide [7], one of the two main endocannabinoids. CBD seems to be a non-intoxicating substance which shows anxiolytic, antipsychotic, anti-inflammatory, anti-oxidative, anti-convulsant and neuroprotective effects [14,15]. CBD is also considered to mediate many of the adverse psychotropic effects of THC, although this research is still emerging [16].

The main challenge for the medical use of cannabinoids is the development of safe and effective methods of use that lead to therapeutic effects, but avoid adverse psychoactive effects. Thus, the first goal is to define what compounds should be used, in what proportions and what are the optimal dosages of cannabinoids in specific indications. A further problem is the safety of prolonged treatment due to the lack of such studies. Generally, there is little high-quality evidence to guide clinicians on the safe use of cannabinoids in symptomatic treatment [17]. Only case studies and small clinical trials report beneficial effects of using medical marihuana.

While we await clinical trials which would help us to define the therapeutic potential of these compounds, it might be useful to analyse the specific clinical situations in which cannabinoids helped to relieve difficult to treat symptoms, such as chronic pain. The authors selected these cases from among their patients treated with cannabinoids—as examples of their use in daily clinical practice in selected indications.

## 2. Case 1: Patient Suffering from Chronic Mixed Pain and Abusing Transmucosal Fentanyl

A 49-year-old male patient suffering from cutaneous T-cell lymphoma (mycosis fungoides, MF) and intercurrent massive ulceration of the left shin. MF type II was diagnosed in October 2014, treated with allotransplantation of hematopoietic stem cells from the peripheral blood in February 2017. Subsequently, the patient received donor lymphocyte infusions (last in April 2018). The ulceration on the shin has been persisting for more than 3 years, recurrently improving and again progressing. The skin lesion is accompanied by severe pain with somatic, inflammatory and neuropathic components.

Additionally, the medical history revealed coexistent long-standing psoriasis, which caused multiple hospitalisations in the past. The patient was treated with methotrexate, cyclosporine, PUVA therapy, adalimumab (12 injections in 2008/2009), ustekinumab (2 pulses). The biological treatment was terminated due to infiltration within the left armpit. A biopsy of the retrieved lymph node led to diagnosing B-cell lymphoma at that time. The patient underwent chemotherapy in the Oncology Centre in 2011.

Moreover, in 2013 the patient was diagnosed with a malignant testicular tumour—teratoma and treated surgically with adjuvant chemotherapy. The patient was also born with phocomelia of the right upper limb (underdeveloped arm, vestigial bones of the forearm).

A tissue sample from the ulceration on the shin, taken in April 2019, was again submitted to surgical pathology for examination in order to verify the diagnosis. The result suggested T-cell lymphoma, although the sparse sample did not allow to determine the type further. In September 2019 the patient received ambulatory radiation therapy using photons X 6MV aimed at the infiltration area of the shin, to the total dose of 40 Gy, with temporary local improvement.

In May 2018 the patient was referred to the Palliative Care Clinic due to very severe pain with somatic, inflammatory and neuropathic components. The pain varied in intensity and was poorly controlled. Paroxysmal episodes of breakthrough pain were assessed as 10 of 10 by Numerical Rating Scale (NRS). The patient described them as pinching, burning, pricking, numbing and electric-like sensations both within the neoplastic ulceration and numerous lesions—during the most intense period the patient had 85 skin lesions and 1 several centimetres long cutaneous horn. During the breakthrough pain episodes, the patient “was sitting by the window in the kitchen and crying because of the pain”.

Before the visit at the Palliative Care Clinic, the patient was treated with oxycodone + naloxone CR tablets—20 mg + 10 mg twice a day and morphine IR in tablets—10–20 mg three to four times a day with poor therapeutic effect. No co-analgesics were used before that time. The treatment was modified by starting duloxetine—30 mg in the morning and transmucosal fentanyl for the breakthrough pain. During the next control visit the patient assessed the therapy as effective and added that he “does not cry by the window because of the pain anymore”.

Due to the severe pain accompanying the advanced skin lesions, the patient was hospitalized twice—in March 2019 and June 2019 at the Palliative Care Department. The massive neoplastic ulceration on the left shin was cleansed of the majority of the necrotic tissue, therefore eliminating the unpleasant smell and resulting in local improvement. The intensification of pain led to non-major modification of the treatment using strong opioids (oxycodone and methadone). Breakthrough pain was treated with transmucosal fentanyl. There was no effect observed after using gabapentin or pregabalin.

In January 2020 the patient was admitted electively to the Palliative Care Department in order to modify analgesic treatment and start cannabinoid therapy, aiming to mitigate inflammation and pain (CBD), as well as reduce excessive use of strong opioids—majorly transmucosal fentanyl in an intranasal form (THC). The therapy consisted of using 15% CBD oil, initially in the dose of 1 drop two times per day (15 mg) and gradually increasing the dose to 2 drops two times per day (30 mg). Moreover, the patient had titration using hemp drought (lemon skunk) containing 19% THC in vaporization—initially used in the dose of 100 mg and afterwards 200 mg. The product was administered at first every 4–6 h, together with subsequent 2–4 additional inhalations every 15 min until the desired effect was achieved.

The therapy led to reducing daily opioid use by 120–200 mg expressed as morphine equivalent doses (including additional doses of intranasal fentanyl). More specifically, methadone (0.1%) was reduced from 18 mg to 15 mg three times per day. Oxycodone/naloxone was reduced from 60 mg/30 mg twice per day to 40 mg/20 mg twice per day on the sixth day of vaporization. Intranasal fentanyl (200 μg per dose) was reduced from 8 doses per day before hospitalisation to 4 doses per day and subsequently to 2 doses per day. Followingly, the product was changed to fentanyl 100 μg per dose (as needed) on the tenth day of the treatment and finally, to fentanyl 100 μg in the form of sublingual tablets on the fourteenth day. Duloxetine (60 mg per day) was sustained.

At the end of the titration, the pain control was achieved concurrently with reduced use of opioids, but without any evident influence on periodical intensifications of pain—episodic and procedural pain (while changing the dressing).

The decision to terminate the treatment using THC in vaporization was made together with the patient, taking under consideration relatively poor impact on the basic and episodic pain, as well as high costs of the therapy. Administration of CBD was maintained for a prescribed period of time due to the immunomodulatory and anti-inflammatory effects.

Currently, although the CBD treatment has been discontinued, the patient does not exceed the previously established opioid doses.

## 3. Case 2: Patient Suffering from Severe Depression under Observation for Affective Bipolar Disorder

A 38-year-old male patient was diagnosed with depression and cross addiction. On the day of admission to the in-patient psychiatric department in 2019 he presented negative toxicological tests, although admitted to drink regularly alcohol (one beer every day) and had a history of using psychostimulants over a dozen years before the hospitalisation. Two months after discharge from the hospital, he decided to stop the recommended treatment himself as, according to his opinion, his state “came back to normal”.

In June 2020 he visited the Psychiatric Clinic accompanied by his wife and mother. For the previous month, he had changed his behaviour, became quiet and isolated himself from others, and his loss of concentration at work (as a forklift operator) was noticed by his colleagues. In addition, he suffered from insomnia and his wife reported his alcohol intake—one or two beers per day. He was diagnosed with a moderate depressive episode and non-organic sleep disorder. The psychiatrist prescribed venlafaxine 150 mg in the morning (as prolonged-release capsules) and agomelatine 25 mg tablets before sleep and wrote out a certificate of incapacity to work.

During the first month of the treatment, the patient reported a partial improvement in mood and psychomotor activity; however, there were still executive deficits and recurrent attention deficits. The patient’s wife noticed that he could be shouted at and not react for several minutes. The treatment was modified by adding lamotrigine starting from 50 mg up to 200 mg daily. The patient reported progressive improvement in general well-being, but according to his wife—there were better and worse days. Although problems with concentration remained, they did not last as long and happened less frequently.

In September 2020, due to pruritus and anxiety as the potential effects of increasing the dose of lamotrigine, the drug had to be withdrawn. Although there were still mild concentration deficits and mood swings, the patient inquired about returning to work. The venlafaxine therapy was stopped and changed to vortioxetine 10 mg orally in the morning. Administering agomelatine was sustained.

In November 2020, the patient complained of progressive fatigue and difficulties in concentrating at work, persistent sadness, feeling helpless, poor sleep interrupted by nightmares and sometimes even suicidal thoughts. He came under observation for affective bipolar disorder. At that time he was diagnosed with a severe depressive episode, without concurrent psychotic symptoms. The certificate of incapacity to work was issued. The therapy was changed by withdrawing vortioxetine and introducing sertraline 100 mg orally in the morning, but sustaining agomelatine. Cannabis flos containing 20% THC was added to the treatment in the dose of 200 mg before sleep in vaporization.

During the following visit the patient reported a significant improvement in the quality of sleep, mood and activity—both executive and cognitive. The improvement had already been noticed on the fourth day after the modification of the treatment and remained stable. The patient reported that he was finally feeling “himself”, for the first time in a several years. Once again he asked to return to work. The treatment remained unchanged.

In March 2021 the patient declared a stable improvement in the quality of sleep, mood, psychomotor activity, concentration and operative memory. Sertraline and agomelatine were withdrawn from the treatment. The patient continued vaporizations of Cannabis flos containing 20% THC in the doses of 150–300 mg daily. Additionally, he revealed that he had stopped smoking cigarettes and drank beer only sporadically.

During the following visit the patient reported that he continued regular Cannabis flos vaporizations, although he reduced the dose to 150 mg per day. He still did not smoke cigarettes or drink alcohol. Moreover, the patient gained 20 kg weight, but he connected it to withdrawing nicotine, as approximately 15 years ago he gained 30 kg after quitting cigarettes. The patient is very satisfied with the effects of the treatment—he sleeps well, his mood is stable and his memory improved. The patient was diagnosed with a past severe depressive episode. The observation for affective bipolar disorder was sustained.

## 4. Case 3: Patient Suffering from Migraine

A 46-year-old female patient suffered from migraine since she was a 19-year-old medical student. During the first year the number of attacks, which usually appeared as left-sided hemicrania, increased from several to 12–14 per month. The patient underwent full radiological diagnostics, which excluded organic causes of the headaches. Menstruation, physical and psychical fatigue emerged as the exacerbating factors—the patient was later actively working as a doctor and was on night duty approximately 8–10 times per month. The headaches also increased their frequency after starting hormonal contraception, which led to its termination after 2 months. The medical history revealed that in the past, in case of less intense headaches (NRS 6/10) the patient used high doses of non-steroidal anti-inflammatory drugs (NSAIDs), such as ibuprofen 400 mg (up to 4 times per day), naproxen 500 mg or diclofenac. In case of severe, intense pain (NRS 9/10) the patient often combined NSAIDs and/or metamizole together with dexamethasone 8 mg per day intravenously. Due to financial reasons, the patient started using sumatriptan instead of other methods in the following years. Right-sided headaches occurred very rarely, but they were much more severe (NRS 10/10) and reacted only to eletriptan combined with steroids (20 mg of prednisone) and NSAIDs. At the time right before starting cannabinoid treatment, the migraine attacks occurred approximately 15 days per month.

The cannabinoid therapy began with a CBD Full Spectrum oil containing 5% of CBD and 5% of cannabigerol (CBG) in the dose of 2–3 drops per day. In the first month the number of migraine attacks decreased to 4. Additionally, the pain was assessed as less intense.

In the following months the patient started to skip the successive doses. She often used only 2 drops of the oil in the evening. The number of migraine attacks stabilized at the number of 4 per month.

When the patient spent time in Greece on vacation, she did not bring the oil with her as she was not certain of the legal aspects of CBD possession there. After two weeks of the vacation without any symptoms, the patient returned to work at the hospital. During the first night duty, she suffered from an intense right-sided headache (NRS 10/10) and nausea. There was no pain relief after using eletriptan, steroids, NSAIDs and the CBD oil. The next day, the patient received cannabis drought containing 20% of THC in vaporization. The pain subsided after 15 min.

Currently, the patient uses CBD and CBG in a stable, titrated dose and THC in vaporization in case of pain.

## 5. Case 4: Patient Suffering from Trigeminal Neuralgia

A 65-year-old patient after dental surgery began to experience severe neuralgia of the right trigeminal nerve. A blockade of the maxillary nerve (V2) was performed on the 6th day. After anesthesia, numbness and stiffness appeared in the innervation of branches V2 and V3 (mandibular). Moreover, intense localized pain made it impossible to speak and eat. The patient reported to the emergency room with pain and increased blood pressure of 208/103 mmHg. The neurological examination revealed swelling of the right cheek, the most intense within the V2 branch, slightly smaller within the V3 branch. Otherwise, there were no focal, meningeal symptoms. Computer tomography of the head was performed and revealed no fresh bleeding, but detected visible hypodense changes in the right internal capsule—probably ischemic with minor calcifications in the right lenticular nucleus. At that time, gabapentin was recommended—initially 100 mg 3 times a day, increasing the dosage every second day to 300 mg, and then finally to 600 mg 3 times a day. During that time, the patient was also taking tramadol, ibuprofen, and sleeping pills as needed, and benefited from rehabilitation.

After 3 years of the treatment, the patient did not improve. There were bouts of drooling, numbness of the face in the V2 area, as well as pain around the temples and upper teeth. She applied to qualify for cannabis therapy. The study showed the same image as before. The patient, who had been suffering for 3 years, had problems with sleeping, any changes in climate during trips ended with salivation and severe pain. Speech and eating also caused pain. The maximum pain on the NRS scale was 10, with tears in her eyes and visible signs of suffering on her face.

The patient began the inhalation therapy of THC 19% and CBD < 1% by vaporization of dried hemp (200 mg/day). Initially, it was recommended to take 1 breath 3 times a day. After one week, the dose was increased to 2 breaths 3 times a day, and after another two weeks—4 breaths 3 times a day. Additionally, CBD oil 10% was added to the treatment by sublingual administration—initially 1 drop 2 times per day (10 mg/day), increased every 3 days by 1 drop.

A follow-up visit after a month revealed the patient’s improved mood and good pain control. Sleep disturbances were resolved. Initially, after each vaporization tachycardia persisted for 15 min, but after 2 weeks the heart rate stabilized, and the patient herself reduced the dose of gabapentin to 100–100–300 mg. It was recommended to continue vaporizations—5 breaths 3 times per day.

Two months after starting the therapy, by the second visit, the patient was taking 3 drops of CBD 10% 2 times per day (30 mg/day), together with 5 or 6 puffs of vaporized THC 19% and CBD < 1% (200 mg/day) as needed. There was a slight dizziness on dose escalation which subsided after 2 weeks. Heart rate remained unchanged before and after vaporization.

Five months after starting the therapy the patient was taking 5 drops of CBD 10% under the tongue 3 times per day (75 mg/day) and 5 inhalations of vaporized THC 3 times per day. The patient reported no pain or problems with sleep. When trying to reduce CBD, numbness appeared on the right side of the face. She was not taking gabapentin at that time and stopped taking all medications that she had been taking up to that point.

A follow-up visit one year after starting cannabis therapy revealed that the patient was satisfied with the results of the therapy, did not feel pain, slept all night and did not report any side effects. It was recommended to maintain the dose.

## 6. Discussion

Numerous clinical trials conducted throughout the last few decades in order to determine therapeutic beneficial effects of cannabinoids led to interesting discoveries, which present a strong relation in treating substance abuse, chronic pain, depression, migraine, neuralgia, and insomnia [11,18,19]—Table 1.

Modern therapeutic challenge in patients suffering from cancer is treating severe, chronic pain, often with neuropathic and inflammatory components. Available methods include administering opioids, which are effective in pain management, but are not deprived of serious side effects and involve the risk of abuse or dependence in prolonged treatment [22]. The United States alone faces a major crisis due to the high prevalence of opioid use disorder, and concurrent overdose deaths [23]. In the face of the currently available literature, cannabinoids may become an alternative for opioids and co-analgesics in pain management, including neuropathic pain [24,25,26,27,28], as it was presented in the example of patients 1 and 5. Although analgesics, antiepileptic drugs and surgical options may be beneficial in the treatment of trigeminal neuralgia, in some cases the therapy is insufficient and requires an alternative approach [29]. Evidence suggests the role of cannabinoids in alleviating neuropathic pain and hyperalgesia by inhibiting neuronal transmission in pain pathways [29]. Therefore, THC and CBD seem to be promising targets in clinical management of trigeminal neuralgia. The most beneficial effects are often observed in the combination of CBD and THC. To increase the anti-inflammatory and anti-euphoric, CBD can be administered first, as it occurred in the case of patient 1.

Moreover, if necessary, combining analgesic treatment with cannabinoids leads to the cumulation of the analgesic effect, which opens the possibility to reduce doses of administered opioids to levels previously ineffective, as well as minimize associated side effects. The additive analgesia in using cannabinoids combined with opioids is explained by studies describing convergent neurochemical mechanisms and neurobiological properties, suggesting a common underlying mechanism of action [30,31]. Additionally, CB2 receptors seem to interact with the opioid system, which may be beneficial in treating inflammatory pain. The anti-inflammatory effect related to CB2 receptor activation causes a downregulation of proinflammatory cytokines and an increase in anti-inflammatory cytokine levels [21], therefore it limits the release of the inflammatory factors stimulated by activation of μ-opioid receptors (MOR) in microglia, which counteracts the opioid tolerance development. This theory has been confirmed in animal studies and clinical trials, in which administering medical marihuana proved to be beneficial during opioid treatment [32].

Many patients noted the positive influence of medical marihuana on withdrawal symptoms compared to prescription drugs, such as opioids, antidepressants or anti-inflammatory medications. Many of those medicines cause serious adverse effects, especially if used regularly for a longer period of time [33]. Moreover, palliative care patients treated with strong opioids often fear drug addiction and undesirable side effects [34]. It is especially significant as millions of patients suffering from cancer pain have opioids prescribed each year [35]. In the first findings from 1889, authors called attention to spectacular results of using marihuana in addiction therapy, which led to relief of persistent symptoms followed by substance use termination [36]. Opioid dependence and withdrawal appear to be subject to the influence of cannabinoids [37]. In the case of substitution therapy, using cannabis in treatment of patients with alcohol or opioid use disorder caused positive results, including reduced use of the substances, decrease in symptoms of depression, improved eating habits, together with decrease in hostility and violence [33]. The effects were observed in patient 2, who stopped smoking cigarettes and significantly limited the consumption of alcohol and patient 1, who was able to reduce the doses of administered opioids.

CB1 receptors and μ opioid receptors (MOR) are expressed in many common areas in the CNS, providing morphological basis to explain the interaction between opioids and cannabinoids in reward and withdrawal. The influence of the receptors on mutual functioning confirms the upregulation of CB1 receptors in the reward pathway in patients using opioids, which seems to have a significant impact on the rewarding properties of opioids and vice versa [28,38,39]. A double-blind randomized placebo-controlled trial assessed the influence of CBD on acute, short-term and protracted withdrawal symptoms in individuals with a heroin use disorder. Compared to the placebo, CBD reduced craving and anxiety related to withdrawal, without any serious side effects [40]. One retrospective analysis examining the relation between cannabis use before and during methadone-based substitution therapy showed that cannabis intake results in relief of withdrawal symptoms and reduction in administered methadone doses [41]. Despite convincing proof from observational studies, clinical data remain unclear [42,43].

In recent years, a new hypothesis has been put forward, implying that all people have a primary endocannabinoid profile, which reflects endocannabinoid levels—anandamide and 2-arachidonoylglycerol, their production, metabolism and receptor expression [44]. The system’s failure leads to circumstances conducive to development of disorders, such as migraine, fibromyalgia, and irritable bowel syndrome. The hypothesis casts light on diseases, which, until recently, were considered to be psychosomatic, burdened with a reputation of disorders resistant to known therapeutic methods [44]. The thesis does not exclude the generally recognized theory that migraine pain is caused by lowering the threshold of nociceptive signal processing in response to release of proinflammatory factors [45]. In the literature there are more and more reports showing the effects of using cannabinoids in migraine treatment, including reduction in pain, nausea and vomiting, together with mitigation of inflammatory response [45,46,47,48].

It is suspected that analgesic properties of cannabinoids may rely on multidirectional activity in pathways—glutamatergic, GABA-ergic, serotonergic, opiatergic and inflammatory [49], raising the pain threshold as an effect [48]. A significant role is played by both CB1 receptors, inhibiting GABA-ergic and glutamatergic transmission, and CB2 receptors, presenting anti-inflammatory properties [21]. It is interesting that triptans, drugs often used to relieve migraine attacks—as in the case of patient 3, except for action in the serotonergic pathway [50], also seem to activate the cannabinoid system [48].

There are reports indicating that treating chronic migraine with medical marihuana emerged to be the only applicable method in patients, for whom the standard and commonly used pharmacological methods were ineffective or could not be implemented for various reasons [48]. Such a situation was observed in the case of patient 3. Moreover, patients who responded well to cannabis treatment were able to reduce doses, or even resign from using prescription drugs, most often opioids [51,52].

There are reports proving beneficial effects of cannabis in treating depression in patients, who did not respond to standard pharmacological methods [53]. Attenuation of signaling in the endocannabinoid system correlates with psychiatric disorders development, such as anxiety, depression and schizophrenia, which has been repeatedly proven in animal studies and clinical trials [54]. The complex pharmacology of several cannabinoids is related to the modulation of release of neurotransmitters engaged in depression development in the serotonergic [55], glutamatergic [56] and endocannabinoid pathways, together with alterations on molecular and cellular levels [57,58]. Experimental animal studies indicate that lower doses of CB1 receptor agonists mitigate anxiety, similarly to antidepressant drugs. Agonists of CB1 receptors seem to increase central transmission of neurotransmitters, such as serotonin and noradrenaline [59,60,61].

One of the observational studies based on assessment of patients using medical marihuana showed a significant decrease in symptoms related to depression, anxiety and stress in the short-term. Variable results depended on dosage, THC and CBD concentrations in the inhaled product, with the highest efficacy coming from the combination of high levels of CDB with low levels of THC in mitigating depression symptoms [62]. Interestingly, in the case of patient 2, using high levels of THC and low levels of CBD proved to be effective in treating depression. The differences bring attention to the need for further research and clinical trials, which would enable to systematize the knowledge regarding cannabinoid dosage in treatment of the aforementioned disorders.

Authors of the study additionally advise caution in the long-term cannabis use, due to alternations in the endocannabinoid system, which paradoxically may increase patients’ susceptibility to depression and reverse the positive effects of the therapy [62]. Moreover, a number of participants enrolled in other trials experienced negative results of the treatment, such as paranoia, irritation, dysphoria, demotivation or depression [11,53,63,64,65,66]. Additionally, the abuse risk assessment is necessary, similarly to the case of using opioids. Elderly people, emotionally stable, who previously rarely abused any kinds of substances, are less likely to develop drug dependence [67,68,69].

Nevertheless, limited evidence derived from clinical trials suggest that using cannabinoids, including prescribed products containing THC, may become an alternative in treatment of anxiety and depression [26,53,57,62,70,71,72,73,74,75,76].

## 7. Summary

Evidence derived from observational studies suggest that using cannabis may help to reduce symptoms, alleviate the course of many diseases, as well as withdrawal symptoms in substance abuse disorder, such as opioid abuse and dependence. The endocannabinoid system undoubtedly plays a vital role in the modulation of functioning of many systems, but further observations and clinical trials are necessary to assess both efficacy and dosage of cannabinoids in certain disorders. Unfortunately, so far there is still not enough clinical data, which would enable us to draw credible conclusions and establish standardized doses in the selected disorders. Every patient should be approached individually with careful assessment of their condition and treated according to the “start low, go slow” principle in order to determine the lowest effective dose. In the series of presented cases cannabinoids were not used as a first-line therapy, but proved their efficacy as a complementary or alternative approach when other available treatment methods did not deliver expected and satisfactory results. Prospective approach to using cannabis in everyday clinical practice, devoid of bias and apprehension on the physicians’ part, aims to study the research and other countries’ experience, where both plant form and pure extract already have medical usage. Although today it may seem unlikely, in the near future cannabis may become widely accessible and remarkably beneficial for our patients.

## Figures and Tables

**Table 1 biomedicines-10-01862-t001:** Effects of CBD and THC [20,21].

Effect	CBD	THC
analgesic	X	X
anti-depressant and mood regulating	X	X
anxiolytic	X	
anti-insomnia		X
anti-inflammatory	X	X
immunosuppressive	X	X
stimulates appetite		X
anti-spasmodic	X	X

CBD, cannabidiol; THC, Δ9-tetrahydrocannabinol.

## Data Availability

Not applicable.

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
