# Peer review of "Cannabinoids—Perspectives for Individual Treatment in Selected Patients: Analysis of the Case Series"

_biomedicines, 2022, doi:10.3390/biomedicines10081862_

Round 1

Reviewer 1 Report

The authors collected interesting data on patients treated with cannabis products to treat different states that did not respond to the approved drug treatments.  Timely paper providing new insight on how to use medical marijuana in patients with neuropathic pain, depression, and migraine. Overall the manuscript is well written but minor editing will improve the quality.

- I am not sure why the authors submitted the manuscript as a review, since they are analysing some patients' cases not reported in the literature before, and they are not reviewing exactly clinical cases in literature either. 

- In general the authors tend to use a few references, they should add the right references to cite the scientists who made the discoveries they are talking about (e.g. line 60 citation 5  is a paper describing only one of the ways how CBD acts, even if in that paper they are talking about the other ways too, the authors need to cite the people who discovered that CBD acts on the metabolism and reuptake of CBD). Additionally, in the discussion (line 355) they stated there is a new hypothesis that all people have a primary endocannabinoid profile, but they don't provide any reference.

- the authors should include why they selected these cases. Were these the only cases available? Were others excluded, if yes why? 

- the authors should collect information about similar clinical cases treated with medical marijuana and try to compile in the summary or in a table some kind of summary of doses used in each case of THC and CBD respectively. Some info are scattered in the manuscript but it would be of great value to see a table summarizing how those specific patients in those specific "state" responded well to medical marijuana. Table 1 could be replaced with one with such information and all the literature related to it. 

- line 463 (reference #14), the author's name has only initials.

Author Response

Thank you very much for a favorable and thorough evaluation of the manuscript. The comments are of great value and we think the implemented alterations will benefit the paper and increase its quality. We hope we have addressed the suggestions correctly and sufficiently.

  1. We agree that the manuscript should be published as a case series rather than a review. Unfortunately, in the submission process we were not able to find this option. We are prepared to change this if the editor allows it. We have already contacted the editor about this issue.
  2. We have added the correct references, which cite the authors of the discoveries, as the reviewer suggested (the changes are marked): lines 41, 42, 44, 47-49, 60-61, 388, 406.
  3. We have provided the missing reference in the discussion (line 359) to the statement about the primary endocannabinoid profile.
  4. The aim of the study was to show the spectrum of indications for exploiting the potential of cannabinoids, not to show a series of cases for the same indication. We have added a statement: “The authors selected these cases from among their patients treated with cannabinoids - as examples of their use in daily clinical practice in selected indications” (lines 76-78).
  5. As much as we would like to provide a table summarizing the doses used in specific indications based on other cases, literature still offers only scarce data on the subject and highlights the importance of treating every patient individually, according to their needs and condition. In order to follow the reviewer’s suggestion, we have added a statement addressing this issue: “Unfortunately, so far there is still not enough clinical data, which would enable us to draw credible conclusions and establish standardized doses in the selected disorders. Every patient should be approached individually with careful assessment of their condition and treated according to the “start low, go slow” principle in order to determine the lowest effective dose” (lines 419-423).
  6. We have added the author's full name in the line 486, reference 22 (previously line 464, reference nr 14).

Reviewer 2 Report

This is a series of case reports on the use of cannabinoids for medical indications such as migraine, depression, severe chronic pain and neuralgia.

It is evident from those reports that cannabinoids may be considered as potential complementary or alternative approach as far as the other more conventional therapeutic management options are exhausted and uneffective.

Authors put their observations in the overall critical context of the ongoing debate about the medical use of marihuana.

Author Response

Thank you very much for a favorable and thorough evaluation of the manuscript. The comments are of great value and we think the implemented alterations listed below will benefit the paper and increase its quality. 

  1. We agree that the manuscript should be published as a case series rather than a review. Unfortunately, in the submission process we were not able to find this option. We are prepared to change this if the editor allows it. We have already contacted the editor about this issue.
  2. We have added the correct references, which cite the authors of the discoveries, as the reviewer suggested (the changes are marked): lines 41, 42, 44, 47-49, 60-61, 388, 406.
  3. We have provided the missing reference in the discussion (line 359) to the statement about the primary endocannabinoid profile.
  4. The aim of the study was to show the spectrum of indications for exploiting the potential of cannabinoids, not to show a series of cases for the same indication. We have added a statement: “The authors selected these cases from among their patients treated with cannabinoids - as examples of their use in daily clinical practice in selected indications” (lines 76-78).
  5. As much as we would like to provide a table summarizing the doses used in specific indications based on other cases, literature still offers only scarce data on the subject and highlights the importance of treating every patient individually, according to their needs and condition. In order to follow the reviewer’s suggestion, we have added a statement addressing this issue: “Unfortunately, so far there is still not enough clinical data, which would enable us to draw credible conclusions and establish standardized doses in the selected disorders. Every patient should be approached individually with careful assessment of their condition and treated according to the “start low, go slow” principle in order to determine the lowest effective dose” (lines 419-423).
  6. We have added the author's full name in the line 486, reference 22 (previously line 464, reference nr 14).
